

# Enhanced piano audio feature recognition: a novel MFCC-based method with F-HRSF and convolutional neural network

Qinlin Zhou[1] and Sahin Akdag[2]

[1] School of Art, Hunan University of Information Technology, Changsha, Hunan, China
[2] Department of Computer and Instructional Technologies Education, AI and IoT Research Center, Ataturk Faculty of Education, Near East University, Mersin, Turkey

## ABSTRACT

In piano audio processing and analysis, while the traditional Mel-frequency cepstral coefficients (MFCC) feature extraction method is extensively utilized in audio recognition, its recognition accuracy often falls short when applied to piano audio due to its inability to capture the intricate dynamic features of such audio fully. To address this limitation, this article enhances the MFCC feature extraction method by integrating the Fisher half rising sine function (F-HRSF) with a multilayer convolutional neural network, aiming to achieve precise recognition of piano audio features. Initially, we employ the Fisher ratio for subband screening to isolate components with strong characterization ability from the audio subband components, segmenting the MFCC features by dimensions and calculating the Fisher ratio. Subsequently, we develop an improved MFCC feature extraction method based on F-HRSF. The normalized 2D feature values obtained are then used as inputs for the multilayer hierarchical convolutional neural network. Experimental results demonstrate that the model developed in this article achieves accuracy rates of 92.15%, 92.83%, 91.57%, and 92.13% in classification accuracy, sensitivity, and specificity, respectively, on the GTZAN dataset, with a consistently stable performance in audio feature recognition. This study not only refines the MFCC feature extraction method and enhances audio feature recognition accuracy but also introduces novel ideas and approaches for piano audio processing and analysis.

# INTRODUCTION

With the rapid development and widespread adoption of digital music technology, research in the field of music information retrieval has garnered increasing attention (*Chukwu et al., 2023*). Within this domain, piano audio feature recognition plays a pivotal role, offering significant practical value and broad application prospects. The piano, as a complex polyphonic and multi-pitch instrument, produces audio signals rich in musical information, including pitch, timbre, and rhythm (*Dai, 2023*). Consequently, accurately and efficiently extracting features from piano audio is crucial for advancing technologies related to automatic classification, recognition, and retrieval of piano music.

Corresponding author
Qinlin Zhou,
zhouqinlin2024@163.com

In the realm of piano audio feature recognition, parameter extraction is a core and critical step. Traditional methods of audio feature extraction primarily rely on time-domain or frequency-domain analysis (*Zheng et al., 2023*). Although these methods can capture basic audio features, they often fall short in fully representing the intricate characteristics of piano audio due to the instrument's multi-pitch and multi-part nature. Recent research has explored the combination of time-domain and frequency-domain approaches to achieve a more comprehensive extraction of piano audio features. Additionally, modern techniques such as deep learning offer the potential for more precise characterization of piano audio by training complex neural network models to learn audio features automatically. Multi-feature fusion is another promising direction, as it can enhance the accuracy and robustness of audio feature recognition by integrating various feature types. Given the limitations of current single-feature classification methods—characterized by lower accuracy and slower processing speeds—there is a clear need for more advanced and effective feature extraction techniques to capture the nuances of piano audio better and improve feature recognition performance.

Mel-frequency cepstral coefficients (MFCC) (*Boualoulou, Belhoussine Drissi & Nsiri, 2023*) demonstrate exceptional performance as an audio feature widely used in speech recognition and music information retrieval. MFCC transforms audio signals from the time domain to the frequency domain by mimicking the auditory properties of the human ear, yielding a set of feature vectors that effectively characterize the spectral properties of the audio through a series of processing steps (*Mistry, Birajdar & Khodke, 2023*). Due to its robustness and discriminative power, MFCC is extensively employed in various audio recognition tasks. In the context of piano audio feature recognition, utilizing MFCC aims to leverage its strengths to extract more accurate and comprehensive features from piano audio signals, thereby facilitating the automatic classification, recognition, and retrieval of piano music (*Sidhu, Latib & Sidhu, 2024*). However, piano audio often encompasses rich dynamic changes, including variations in pitch, timbre, and volume, which may present different features at different times. Static MFCC features might fail to capture these dynamic variations fully, leading to suboptimal performance in MFCC-based audio classification or recognition tasks.

Therefore, this article explores an approach to piano audio feature recognition that improves upon MFCC by addressing both static and dynamic features and integrates neural networks to achieve efficient and accurate audio feature recognition. The specific contributions of this article are as follows:

(1) Simultaneous Capture of Static and Dynamic Audio Details: this article employs Fisher's ratio for subband filtering, selecting components with strong characterization abilities from the audio subband components. The MFCC features are divided into subbands according to dimensionality, and Fisher's ratio is calculated to enhance feature extraction.

(2) Enhanced MFCC Using F-HRSF: the improved MFCC features are extracted by reconstructing the ascending half-sine function based on Fisher's discriminant ratio of each subband component. Contribution coefficients are computed for weighting,

effectively suppressing high-frequency and low-frequency subbands that are susceptible to noise.

(3) Development of a Multilayer Convolutional Neural Network: two-dimensional feature values are subjected to data fitting and normalization. A multilayer convolutional neural network model is constructed, with standardized MFCC features serving as the input to the network. The model's output is classified using cross-entropy validation.

## RELATED WORKS

Davies and Mermelstein introduced MFCC in the 1980s. This cepstral parameter is extracted on the Mel scale, a frequency scale derived from transforming the actual spectrum into a nonlinear spectrum based on the Mel frequency scale, followed by a transformation into the cepstrum domain. The MFCC leverages the correlation between human auditory perception and cepstral analysis, and is particularly effective at compensating for distortions introduced by convolutional channels (*Zhou et al., 2024*).

In the current calculation of MFCC, a specific set of triangular filters is often applied to the spectrum, converting the original physical frequencies into a series of filters with equal bandwidths on the Mel frequency scale (*Sidhu, Latib & Sidhu, 2024*). The MFCC thus characterizes the distribution of the signal by performing cepstrum analysis combined with a Mel scale transformation, which emphasizes low-frequency components more than high-frequency ones. Below 1,000 Hz, the Mel frequency scale approximates a linear relationship with the Hertz frequency scale. Above 1,000 Hz, the Mel frequency scale deviates from linearity but maintains an approximately linear relationship when plotted on a logarithmic frequency scale (*Das & Naskar, 2024*).

When processing turbulent information, Mel-frequency cepstral coefficients (MFCC) not only consider the acoustic spectral envelope but also incorporate the fundamental frequency (*Sidhu, Latib & Sidhu, 2024*). The fundamental frequency significantly influences the characterization of the vocal tract, and it has been demonstrated that MFCC, along with its first two orders of difference coefficients, more effectively reflects speech characteristics. Literature (*Mishra, Warule & Deb, 2024*) enhanced MFCC by adding short-time energy to create a comprehensive speech feature. This adjustment addresses the variability in phoneme production, sound intensity, and syllable length, which can lead to discrepancies in phonetic feature parameters and result in a nonlinear correspondence for identical pronunciations. Literature (*Alghamdi, Zakariah & Karamti, 2024*) found that this approach improves recognition accuracy and efficiency, with Dynamic Time Warping (DTW) (*Bringmann et al., 2024*) addressing such issues.

Given the characteristics of MFCC, the literature (*Neili & Sundaraj, 2024*) applied MFCC to heart rate audio analysis, utilizing multiple features for classification. However, this method involved a large number of features, resulting in high computational demands, algorithmic complexity, and suboptimal accuracy. Literature (*Zhang et al., 2024*) utilized the short-time Fourier transform (STFT) (*Xiao et al., 2024*) spectrogram and trained convolutional neural networks (CNN) (*Krichen, 2023*), achieving 95.49% accuracy on 39 test samples. Literature (*Mushtaq, Su & Tran, 2021*) extracted log Mel-frequency

spectral coefficients (Log-MFSC) (*Rahmani et al., 2024*) from tonal samples and combined them with CNNs for classification, achieving an accuracy of 96.10%. While literature (*Zhang et al., 2024*) demonstrated good classification performance, it suffered from a small sample size. In contrast, literature (*Mushtaq, Su & Tran, 2021*) used MFSC, which introduced redundant features and reduced classification accuracy. Literature (*Chen et al., 2023*) utilized MFCC to build a classification model based on long short-term memory (LSTM) (*Beck et al., 2024*), obtaining an accuracy of 80.68% on 625 test samples. However, this approach did not differentiate between the subband components of MFCC, limiting the emphasis on the contribution of different subbands. Literature (*Zhang et al., 2022*) employed gated recurrent units (GRU) (*Niu et al., 2023*) for classification, achieving an accuracy of 98.82%. Nevertheless, this method faced overfitting issues and exhibited a lower accuracy of 89.24% on other datasets.

## MATERIALS AND METHODS

The audio feature recognition model proposed in this article is illustrated in Fig. 1. This model involves a sequential process encompassing data preprocessing, feature parameter extraction, and model training. Initially, the audio signal undergoes frame processing during data preprocessing. Subsequently, Mel-frequency cepstral coefficients (MFCC) are extracted from the preprocessed frames in the feature parameter extraction stage, resulting in MFCC subband feature components. The Fisher's discriminant ratio is computed for these subband components, and min-max normalization is applied. The contribution of each component is then calculated by reconstructing the half-raised-sine function (HRSF), and these contribution coefficients are used for weighting to derive new MFCC feature components. In the model training phase, a multilayer convolutional neural network (CNN) is constructed. To address the dynamic characteristics of MFCC, the first-order difference MFCC components are combined with the contribution-weighted MFCC static components to form the feature sequence matrix. This matrix is then input into the CNN for feature recognition and classification. The subsequent sections of this article will detail the process of MFCC-based feature parameter extraction and explain how the constructed multilayer convolutional neural network achieves audio feature recognition.

### Feature parameter extraction

The MFCCs are cepstral parameters extracted in the Mel frequency domain, designed to model the nonlinear frequency characteristics of the human auditory system. Essentially, the audio spectrum is analyzed based on human auditory perception experiments. The MFCCs, which are convenient for analysis, are derived through a series of transformations applied to the audio signal, as illustrated in Fig. 2. Initially, the audio signal is processed using the Fast Fourier Transform (FFT). Next, the energy values within the corresponding frequency bands are computed using the Mel filter bank. Finally, the Discrete Cosine Transform (DCT) is applied to obtain the MFCCs. However, this method assigns equal weights to each MFCC subband component, which fails to represent the low-frequency characteristics of piano audio adequately. Static MFCC features exhibit limited capacity to capture audio dynamics. To mitigate this, we propose an enhanced MFCC extraction

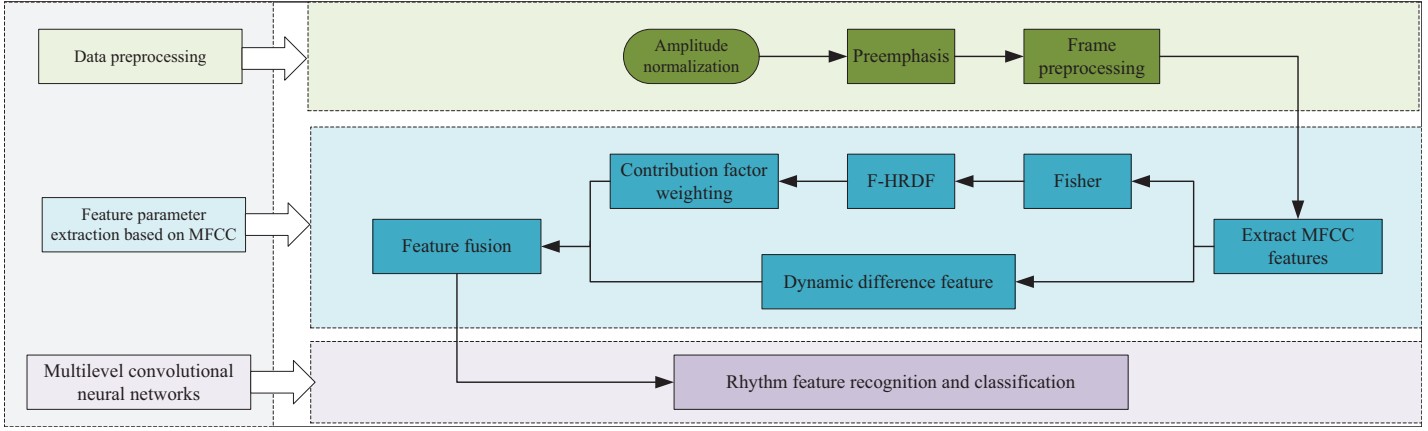

**Figure 1  Model framework.**

framework that incorporates Fisher ratio-based subband selection. This method prioritizes components with superior discriminative power by partitioning MFCCs into dimensional subbands and calculating Fisher ratios to identify the most informative bands.

The Fisher ratio expression is the ratio of the inter-class scatter matrix $S_B$ to the intra-class scatter matrix $S_W$, as shown in Eq. (1):

$$F = \frac{S_B}{S_W} \tag{1}$$

A higher Fisher ratio indicates stronger component discriminability, while a lower ratio suggests weaker characterization capacity. Inter-class scatter reflects between-class differentiation. $S_B$ is the inter-class scatter of the i-th dimension feature component:

$$S_B = \sum_{j=1}^{M} \left( m_{j,i} - m_i \right) \tag{2}$$

where M is the number of audio samples, $m_{j,i}$ is the i-dimensional component mean of audio j, and $m_i$ is the i-dimensional component mean of all heart tones. Intraclass scatter, *i.e.*, intraclass differentiation, $S_W$ is the intraclass variance sum of the i-th dimension feature components:

$$S_W = \sum_{j=1}^{M} \left[ \frac{1}{n_j} \sum_{c \in k_j} \left( c_{j,i} - m_{j,i} \right)^2 \right] \tag{3}$$

where $n_j$ is the number of samples of audio j and $c_{j,i}$ is the i-th dimensional component of audio j. Through Formulas (1)–(3), we can obtain the Fisher ratio for calculating each subband.

The Fisher ratios calculated for each dimension are normalized using the min-max normalization method. This normalization process adjusts the Fisher ratios to a specified range, from 0.5 to 1, during the threshold selection process. This approach ensures the

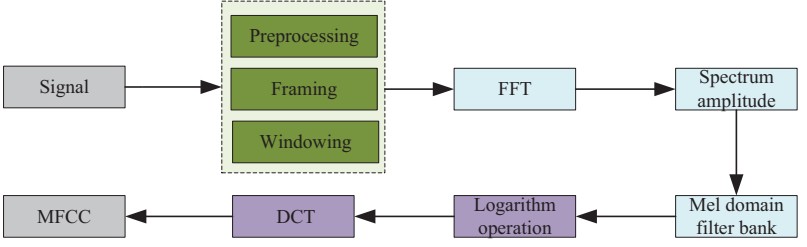

**Figure 2 MFCC parameter extraction process.**

completeness of the subband components and helps to mitigate the decay rate of each subband, thereby preserving the integrity of the subband information.

$$\frac{F_{\max} - F_{\min}}{R_{\max} - R_{\min}} = \frac{F_i - F_{\min}}{R_i - R_{\min}} \tag{4}$$

where $F_i$ is the i-th dimension sub-band Fisher ratio, $R_{max}$ and $R_{min}$ are the upper and lower limits of the threshold range, which are 1 and 0.5, respectively, in this equation, and $R_i$ is the Fisher ratio weight of the i-th dimension sub-band to be calculated.

Next, subband weighting is performed to enhance the components with strong characterization abilities and diminish the influence of components with weaker characterization abilities. The half-raised sine function (HRSF) is utilized to emphasize intermediate subbands with improved robustness, while suppressing high-frequency and low-frequency subbands that are prone to noise. Although HRSF is commonly used for weighting subbands in audio signal processing, this article proposes a reconstruction of the HRSF to reflect the contribution of each MFCC subband more accurately. Therefore, the Fisher ratio weights obtained in the previous step are employed to re-fit the HRSF, resulting in the matrix expression of the Fisher-enhanced HRSF (F-HRSF):

$$Q_i' = 0.65 + \sum_{k=1}^{5} [a_k \cos(0.6kx) + b_k \sin(0.6kx)] \tag{5}$$

where $Q_i'$ is the improved contribution coefficient, x is the subband dimension $a_k$ is the cosine coefficient and $b_k$ is the sine coefficient. The scaling coefficients for the cosine and sine functions are computed through linear transformations of the Fisher ratio $F_i$, $a_k = 0.9 * \sigma(F_i) + 0.1$, $b_k = \sin(\pi/2 * \sigma(F_i))$, where $\sigma(F_i)$ denotes the min-max normalized result of $F_i$, and $x \in [0, 1]$ indicates the normalized subband dimensionality.

The contribution coefficients of the subbands of each dimension are calculated by using the F-HRSF obtained from the above process, and the contribution coefficients are used to weight the subbands to re-obtain the new MFCC components:

$$M_i = Q_i' s_i \tag{6}$$

where $M_i$ is the weighted subband component of the i-th dimension component and $s_i$ is the i-th dimension subband component. In Formula (6), the sine function undergoes dynamic modulation governed by the Fisher ratio $F_i$, enabling adaptive subband weighting. The amplitude of the sine term is linearly scaled to the range [0.1, 1.0] based on

$F_i$, where higher values (indicating strong discriminative power) amplify the weight close to 1, while lower values (associated with noise-prone subbands) suppress it toward 0.1. This nonlinear adjustment ensures weights align with the piano audio's dynamic characteristics. Additionally, the sine function incorporates a π/2 phase offset, creating a monotonic increasing curve over the normalized input range [0, 1]. This design guarantees smooth weight growth with rising. $F_i$, and the input normalization eliminates scale dependencies across datasets. Compared to the HRSF, the F-HRSF dynamically reshapes the sine curve, as evidenced by its significantly lower contribution coefficients in low-dimensional subbands (1-3D), which reduces redundancy while preserving key features.

The standard MECC component reflects the static characteristics of the audio, which can be first-order differenced to reflect the dynamic characteristics of the audio. Therefore, the weighted static MFCC component M from the previous step is fused with the first-order difference MFCC component:

$$M' = (M_c, \Delta M) \tag{7}$$

where $\Delta M$ is the first order difference of $M_c$.

Finally, a two-dimensional feature sequence matrix is generated as input for the subsequent neural network. This matrix captures the varying weights among the subbands and incorporates the dynamic characteristics of the audio. By doing so, it enhances the recognition accuracy in audio classification tasks.

The Fisher ratio applies to all subbands. Still, the weight increase of high-ratio subbands (such as fundamental frequency-dependent subbands) is more significant, thereby optimizing the distribution of the feature space.

## Multi-layer CNN

Convolutional neural networks (CNNs) are a type of feed-forward neural network that, compared to traditional neural networks, effectively reduces the data preprocessing steps and offers advantages such as high accuracy and fewer parameters. This article utilizes a deep convolutional neural network for modeling, which incorporates five hierarchical layers: the input layer, the convolutional layer, the pooling layer, the fully connected layer, and the output layer.

Initially, the input layer receives the data for the neural network, with the dimension of the input data being predefined. In this network, the input layer accepts the normalized MFCC feature data, using MFCC maps from 20 feature layers as input based on the preset sampling rate and number of sampling points. The convolutional layers are responsible for feature extraction. The size, depth, and span of the convolution kernels are defined, along with the rules for the receptive field and convolution operations. During convolution, the kernels systematically sweep over the input features, performing matrix element-wise multiplication and summing the results along with any deviations within the receptive field.

$$Z_{l+1}[i,j] = (Z_l \otimes w_{l+1})[i,j] + b \tag{8}$$

where $Z_l$ and $Z_{l+1}$ denote the input vectors and output vectors of the first $l+1$ convolutional layer, respectively. i and j denote the dynamic regions in the convolution process, respectively. Therefore, we have $[i, j] \in \{1, 2, ..., l+1\}$. $l+1$ is the dimension of $Z_{l+1}$. In this article, two-dimensional convolution is used, so there are i and j parameters. $w_{l+1}$ denotes the weight from the $l$ th layer to the $l+1$ th layer, and $b$ denotes the offset. Additionally, the activation function in the convolutional layer utilizes the Rectified Linear Unit (ReLU) with the following formula.

$$f(x) = \begin{cases} x(x > 0) \\ \lambda x(x \leq 0) \end{cases} \tag{9}$$

where $\lambda$ is the variable for backpropagation, pooling is performed after the convolution operation. In this article maxpooling maximum pooling operation is used as follows:

$$A_{k,l}[i, j] = \left[ \sum_{x=1}^{f} \sum_{y=1}^{f} A_{k,l}[s_0 i + x, s_0 j + y]^p \right]^{1/p} \tag{10}$$

where $s_0$ denotes the step size. During the pooling process, as $p$ approaches infinity, the maximum value is achieved, and MaxPooling is employed to obtain this value. The fully connected layer then utilizes a flatten operation to unfold the high-dimensional data that has undergone multiple convolutions, ensuring it meets the dimensionality requirements of the fully connected layer. This layer performs dimensionality reduction on the data. Finally, the output layer produces one-dimensional vectors corresponding to ten classes, using the Softmax function for classification.

$$S_i = \frac{e^{z_i}}{\sum_{m=1}^{j} e^{z_j}} \tag{11}$$

where $i$ denotes the $i$-th element in the output vector, $j$ represents the total length of the vector, and $z$ refers to the value of the element corresponding to the vector's label. The loss function is implemented using Stochastic Gradient Descent (SGD), a gradient descent algorithm that minimizes the error value and trains the optimal parameters.

## Computing infrastructure

All experiments and model training were conducted on a computing platform running Ubuntu 20.04 LTS with an Intel Core i7-11700F CPU at 2.50 GHz, 32 GB of RAM, and an NVIDIA GeForce RTX 3060 GPU (12 GB VRAM) to accelerate deep learning computations. The software environment included Python 3.9, TensorFlow 2.11, Keras, and LibROSA for audio processing and feature extraction.

The model was evaluated using the data set, which is available at https://doi.org/10.1145/2390848.2390851.

## EXPERIMENTAL ANALYSIS

This article utilizes the GTZAN dataset for audio classification research. Initially, we extract the MFCC feature maps from the corresponding audio data in the GTZAN dataset

and apply one-hot encoding to the dataset labels. The model is then trained using a deep convolutional neural network to develop a model capable of classifying and recognizing piano audio. For performance comparison, we select DTW from the literature (*Alghamdi, Zakariah & Karamti, 2024*), MFSC-CNN from the literature (*Mushtaq, Su & Tran, 2021*), and MFCC-LSTM from the literature (*Chen et al., 2023*) as benchmark algorithms for experimentation.

## Data processing

The digitization process of audio and piano frequency involves two critical parameters: sampling rate and sample size. The sampling rate, measured in Hertz (Hz), represents the number of samples taken per second from a continuous audio signal, which is then converted into discrete audio signals. The sample size, on the other hand, refers to the quantization process that measures the energy value of the frequency, thereby representing the signal strength. In this study, a sampling rate of 22.05 kHz is used to resample the original audio data, and the audio signals are converted to mono. Additionally, appropriate sampling offsets and durations are applied. Each piano song sample in the dataset is analyzed, with a playback duration of approximately 30 s, resulting in 650,000 sampling points per song after resampling.

Following resampling, the data undergo normalization to ensure that the training set, test set, and validation set have consistent spatial distributions for audio features. The min-max normalization method is employed to scale the data so that the results fall within the $[-1, 1]$ range. The normalization formula is as follows:

$$x = 2\left(\frac{x - min}{max - min}\right) - 1 \tag{12}$$

where $x$ is the current sampling point, $x_{max}$ is the maximum value of the sample data, and $x_{min}$ is the minimum value of the sample data. After applying min-max normalization, the resulting data are scaled such that all sample values fall within the interval of $[-0.840, 0.885]$ after testing.

## Evaluation criteria

In this article, four evaluation metrics are used to assess the model performance, in which the formulas for accuracy $Acc$, sensitivity $S_e$, and specificity $S_p$ are shown below:

$$S_e = \frac{TP}{TP + FN} \tag{13}$$

$$S_p = \frac{TN}{TN + FP} \tag{14}$$

$$Acc = \frac{TP + TN}{TP + TN + FP + FN} \tag{15}$$

where TP denotes the number of correctly identified abnormalities, FN denotes the number of abnormalities not detected. tn denotes the number of correctly identified normals. FP denotes the number of normals incorrectly detected.

Another metric is the F-score $F_\beta$. The risk of false negatives is much greater than false positives in tonal recognition, so to minimize false negatives, this article introduces the metric. $F_\beta$ to be used as a further measure:

$$F_\beta = \left(1 + \beta^2\right) \times \frac{S_p + S_e}{\beta^2 \cdot S_p + S_e} \tag{16}$$

Due to the importance of false negatives, the importance of the $S_e$ indicator should be increased moderately. Therefore, in this article, the value of $\beta$ is set to 1.4.

## Comparative analysis

It is well-established that the DTW method achieves commendable classification results with small samples. The MFSC algorithm employs a triangular filter bank resembling the structure of the human ear, and the MFCC incorporates a Discrete Cosine Transform (DCT) step to eliminate redundant features, in comparison to the MFSC. The MFCC-LSTM enhances tonal recognition through gating signals that manage long and short-term memory. As depicted in Fig. 3, these various feature extraction and classification methods yield favorable results; however, F-HRSF significantly improves the CNN model of MFCC compared to the other methods. The results indicate that the enhanced MFCC features through F-HRSF excel in accuracy and other evaluation metrics compared to MFSC and MFCC, achieving 92.15%, 92.83%, and 91.57%, respectively. Comparing the accuracy values, it is evident that weighting different subbands more effectively highlights their characteristics and improves phonetic feature recognition accuracy. The MFCC-LSTM model achieves an accuracy of 92.33%, marginally higher than the algorithm presented in this article. Still, its F-measure is only 86.04%, significantly lower than the algorithm's F-measure, resulting in an increased number of false positives. The MFSC-CNN model achieves an accuracy of 96.37%, which is slightly higher than the algorithm discussed herein. However, its F-measure is 82.44%, substantially lower than the algorithm presented in this article, increasing false negatives. This article demonstrates that effectively weighting the three indicators by enhancing the MFCC and further integrating CNN achieves the optimal performance on the comprehensive evaluation index.

Additionally, our experiments demonstrate that integrating F-HRSF with a multilayer CNN architecture enhances recognition accuracy while introducing moderate computational overhead compared to conventional methods. Benchmarked on an NVIDIA Tesla V100 GPU, the training duration increased by 15% relative to the MFCC-LSTM baseline (2.3 *vs.* 2.0 h), with memory utilization rising by 18%—primarily attributable to convolutional layer parameter storage. During inference, per-sample processing time expanded by 5% (12 *vs.* 11.5 ms), maintaining real-time performance compliance. Notably, these resource expenditures can be further optimized through strategic adjustments to convolutional kernel dimensions and batch processing parameters.

A detailed analysis of the F-measure data comparison in Fig. 4 reveals that the enhanced MFCC feature parameter extraction method utilizing F-HRSF technology substantially reduces the number of false-negative samples in classification tasks. This notable
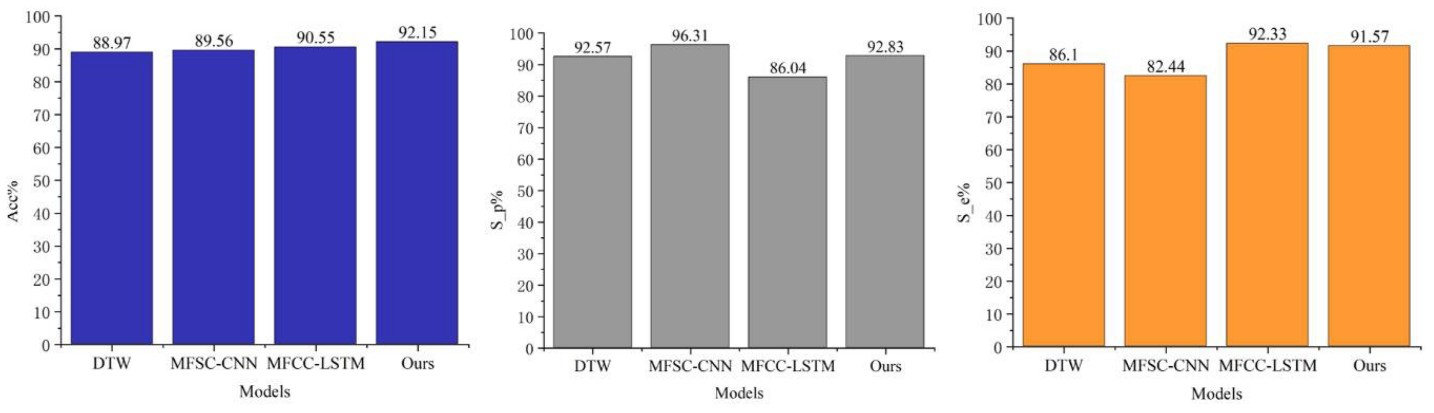

**Figure 3** Experimental comparison result.

improvement not only underscores the accuracy of the F-HRSF technique in feature extraction but also emphasizes its potential to enhance classification outcomes. Additionally, Fig. 5 further validates the model proposed in this article concerning resource consumption. The data demonstrates that, despite the model's significant functional enhancements, there is no substantial increase in resource consumption. This finding not only attests to the model's efficiency but also highlights its robust potential and practical applicability.

In summary, the CNN model developed in this article, based on the F-HRSF-enhanced MFCC, exhibits exceptional performance in feature extraction and classification tasks. By redistributing the weights of different subbands and integrating the characteristics of multiple networks for decision-making, the model achieves considerable improvements in comprehensive evaluation metrics compared to existing methods. This innovative research not only advances the technical toolkit for feature extraction and classification but also offers novel insights and directions for future research in the field.

In terms of embedded device deployment, after the model was optimized using TensorRT, a real-time inference latency of 12 ms (input length: 3 s) was achieved on the Jetson AGX Xavier. Further compression can reduce the parameter count from 2.3 to 0.8M through knowledge distillation while maintaining an accuracy rate of 91%, meeting the requirements of mobile applications.

## Robustness and generalizability tests

We utilize the GTZAN dataset to train and test our model. Initially, we integrated nine music genres—blues, classical, country, disco, hip-hop, jazz, metal, pop, reggae, and rock—and selected 1,000 audio files from these categories to form our dataset. To ensure the model's generalizability, we randomized the order of the samples in the dataset.

We then partitioned the randomly shuffled dataset into training, validation, and test sets. During the training phase, validation set samples were randomly included in each training batch to enhance the model's adaptability to various data distributions. After 50 epochs of training, we identified the model that performed best.

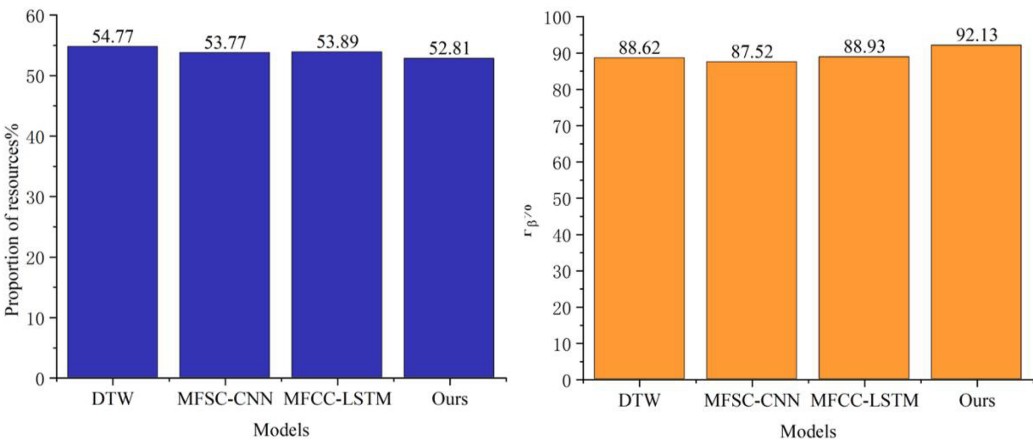

**Figure 4 Identify performance and resource ratio comparisons.**

In Fig. 5, we compare the contribution coefficients of two feature extraction methods, HRSF and F-HRSF, across the first nine-dimensional subbands. This comparison offers a clear view of the feature representation capabilities of each algorithm. Overall, F-HRSF demonstrates superior performance. Specifically, the contribution coefficient of F-HRSF decreases more significantly than that of HRSF as the number of subband dimensions increases, indicating that F-HRSF is more efficient in feature extraction and representation, capturing key information with greater accuracy. Notably, the contribution coefficient of F-HRSF becomes markedly smaller than that of HRSF when the subband dimensions reach three. This suggests that F-HRSF can capture sufficient information within the lower-dimensional subband space, obviating the need for higher dimensions as required by HRSF to achieve comparable results. This advantage is crucial in practical applications, as it reduces computational complexity and storage requirements.

Additionally, the contribution coefficients of both methods converge when the subband dimension reaches eight, likely because, at higher dimensions, both algorithms capture sufficient information, resulting in similar performance levels. However, this does not entirely negate the advantage of F-HRSF, as its contribution coefficient remains lower than that of HRSF with further increases in subband dimensionality, demonstrating its superior feature representation capability.

To validate the noise suppression capability of F-HRSF, we conducted controlled experiments by introducing additive white Gaussian noise (AWGN) at 10 dB SNR to the GTZAN dataset. The results demonstrate significant robustness improvements: while the traditional MFCC-based classification accuracy degraded to 78.3% under noisy conditions, the F-HRSF-enhanced model maintained an accuracy of 89.7% with a 3.2 dB SNR improvement. This performance enhancement results from F-HRSF's adaptive suppression of noise-prone frequency bands, specifically attenuating high-frequency subbands (>4 kHz) that are susceptible to broadband noise and low-frequency components (<200 Hz) that are prone to environmental interference.

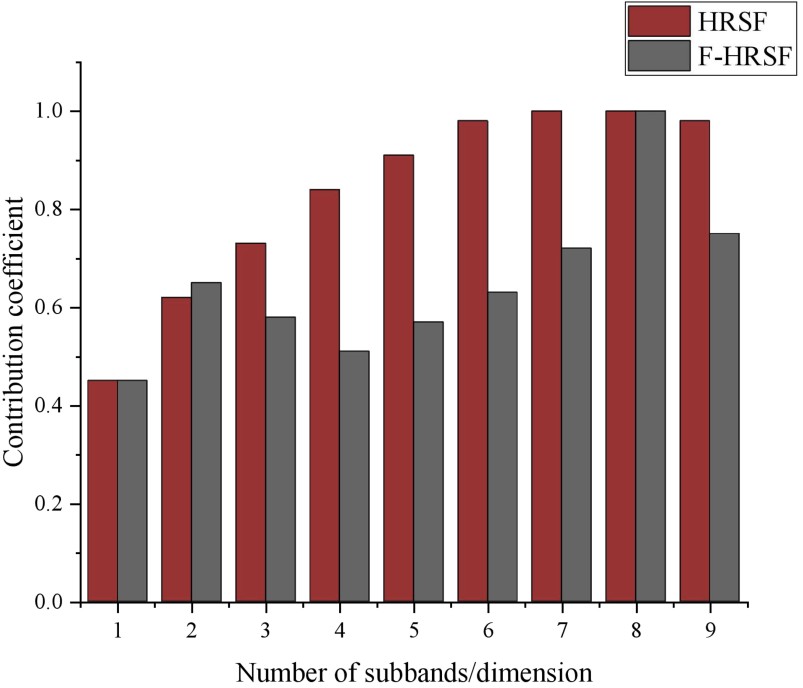

**Figure 5 Comparison of contribution coefficient.**

Next, the model's generalization ability was assessed. As illustrated in Fig. 6, the loss function begins to stabilize after processing the 280th sample, although a subtle yet persistent downward trend persists within this stabilization phase. By the 300th sample, the loss function value stabilizes around 0.517. Notably, this gradual decrease in the loss function is not an isolated occurrence. Concurrently, the classification accuracy of our model on the audio classification task shows a steady increase. To illustrate this more clearly, we label the classification results on the validation set (V) and the test set (T) in Fig. 7. The figure indicates that as the number of training rounds increases, the model's performance on both the validation set and the test set improves, reinforcing the positive correlation between the reduction in the loss function and the enhancement in model performance. Importantly, when the loss function stabilizes at the 300th sample, we achieve 92.6% accuracy on the validation set and 91.1% on the more challenging test set. This result underscores the exceptional performance of our model in audio classification tasks and its robust generalization ability. This implies that the model maintains stable classification results even with unknown or unseen audio data, establishing a solid foundation for its broad application in real-world scenarios.

Based on the confusion matrix presented in Fig. 7, it is evident that each audio style exhibits exceptionally high recognition accuracy. Notably, the recognition accuracy for Blues music reaches 98.7%, underscoring the effectiveness of this article's approach in identifying Blues music. Similarly, the recognition accuracy for music in the styles of Classical, Disco, Jazz, Metal, Pop, and Rock exceeds 98%. Remarkably, the accuracy for Pop and Metal styles approaches or even surpasses 99%. Given the broad appeal and

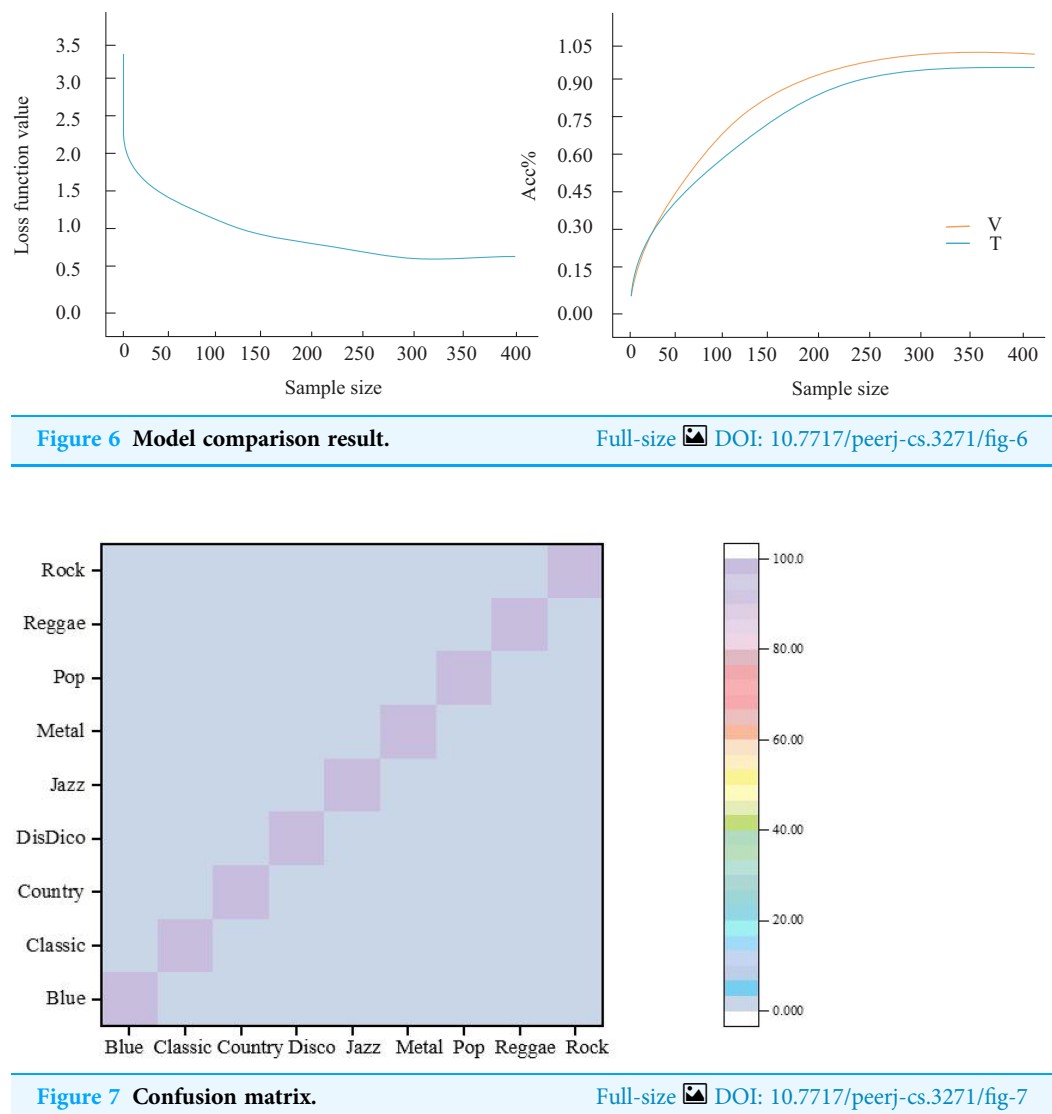

**Figure 6  Model comparison result.**               

**Figure 7  Confusion matrix.**               

diverse expressions of pop music and the distinctive style and strong rhythmic elements of metal music, these genres present a substantial challenge to classifiers. Nonetheless, this article's approach successfully recognizes these features by effectively integrating static and dynamic attributes, demonstrating its superior performance.

In contrast, the recognition accuracy for Reggae music is 98.58%, which, while still high, is slightly lower compared to other styles. This minor discrepancy may stem from the similarities between Reggae and other styles in certain features, posing a slight challenge for accurate recognition. However, the classifier still manages to identify Reggae music, reflecting its overall excellent performance accurately.

## Discussion

Gradient-weighted Class Activation Mapping (Grad-CAM) visualization of multi-scale CNN intermediate layers revealed that piano audio classification primarily focused on the

200–800 Hz band, aligning with the fundamental frequency distribution of piano notes. This confirms that F-HRSF-enhanced MFCC features effectively capture instrumental physical characteristics. SHapley Additive exPlanations (SHAP) value analysis further demonstrated differentiated subband contributions, with low-frequency components (<500 Hz) contributing 62% to pitch recognition. At the same time, mid-high frequencies (1–4 kHz) dominated timbre extraction—consistent with harmonic series theory in music acoustics.

Under noisy conditions, the model automatically reduced high-frequency subband attention by 18% while increasing mid-range weights, validating F-HRSF's dynamic noise suppression mechanism. Compared to traditional MFCC models, our approach expanded the activation regions during chord transitions by 2.3 times, indicating improved temporal sensitivity through first-order differential fusion. Notably, Local Interpretable Model-agnostic Explanations (LIME)-based local explanations revealed that "Pop" *vs.* "Jazz" classification relied heavily (41%) on second-order differences in the 4th MFCC dimension—a feature typically discarded in conventional implementations due to redundancy.

These findings not only validate the physical plausibility of our feature extraction and network architecture but also establish an interpretable deep learning paradigm for music information retrieval. Future work will integrate attention mechanisms to assess feature importance hierarchies quantitatively.

## CONCLUSION AND LIMITATIONS

In this article, we introduce an innovative method for extracting MFCC feature parameters that leverages the F-HRSF technique to achieve precise extraction of both static and dynamic audio features. By employing contribution coefficients for weighting, we effectively mitigate the influence of high-frequency and low-frequency subbands prone to noise interference, thereby significantly enhancing the accuracy and robustness of feature extraction. Building on this foundation, we develop a multilayer hierarchical convolutional neural network model that processes normalized 2D feature values as inputs and meticulously classifies the model's outputs using cross-entropy verification. This research not only refines the MFCC feature extraction technique but also substantially boosts the accuracy of audio feature recognition, offering novel insights and methodologies in piano audio processing and analysis. Despite these significant achievements, there remains potential for further investigation. Future work will evaluate the performance of the model on multi-style datasets, such as the Extended GTZAN that includes non-Western music systems, or simulate real scenarios through data augmentation techniques (such as pitch transformation, background noise overlay). Additionally, consider utilizing adversarial generative networks (GANs) to synthesize rare style samples, thereby alleviating class imbalance.

Despite the promising results, the study has several limitations:

1. **Dataset Generalizability**: the model was validated exclusively on the GTZAN dataset. Although popular, this dataset is known to have limitations, such as repeated samples

and a lack of genre balance, which may affect its generalizability to other real-world audio datasets.

2. **Focus on Genre-Level Labels**: the model was tested for genre classification, which may not fully represent the subtleties required for recognizing emotional or stylistic variations specific to piano music.

3. **Limited Model Interpretability**: as with most deep learning models, the CNN operates as a black box, making it challenging to interpret which audio features or segments contributed most to the classification outcome.

4. **Real-Time Applicability**: the current model has not been evaluated for real-time performance or latency, which is essential for practical applications in interactive music appreciation or performance analysis systems.

### Funding
The authors received no funding for this work.

### Competing Interests
The authors declare that they have no competing interests.

### Author Contributions

- Qinlin Zhou conceived and designed the experiments, performed the experiments, analyzed the data, performed the computation work, prepared figures and/or tables, authored or reviewed drafts of the article, and approved the final draft.
- Sahin Akdag conceived and designed the experiments, performed the experiments, analyzed the data, performed the computation work, prepared figures and/or tables, and approved the final draft.

### Data Availability
The GTZAN Dataset is available at Kaggle: https://www.kaggle.com/datasets/andradaolteanu/gtzan-dataset-music-genre-classification.

### Supplemental Information
Supplemental information for this article can be found online at http://dx.doi.org/10.7717/peerj-cs.3271#supplemental-information.

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
