# Peer review of "Enhanced piano audio feature recognition: a novel MFCC-based method with F-HRSF and convolutional neural network"

_PeerJ Computer Science, doi:10.7717/peerj-cs.3271_

## Round 0.1 · original submission · Major Revisions

Dear authors ,

Thank you for submitting your manuscript titled "Enhanced Piano Audio Feature Recognition: A Novel MFCC-Based Method with F-HRSF and Convolutional Neural Network"

After an initial evaluation by the editorial team and peer reviewers, we have completed our review process. While your work presents a novel integration of the Fisher Half Rising Sine Function with MFCC and convolutional neural networks—an approach that is timely and relevant to piano audio analysis—the manuscript in its current form requires major revisions before it can be considered for publication.

Below is a summary of the key issues raised by the reviewers and editorial board:

AE Comments: If the title was included above or nearby, ensure it clearly reflects the enhancement of MFCC with deep learning for piano audio recognition
The abstract is overly dense and includes long, complex sentences. Consider breaking down ideas into shorter, clearer sentences for better readability
The term “Fisher half rising sine function (F-HRSF)” is not standard or widely known. Briefly clarify its purpose or advantage in feature enhancement.
The sentence “achieves accuracy rates of 92.15%, 92.83%, 91.57%, and 92.13% in classification accuracy, sensitivity, and specificity, respectively” is confusing — four metrics are listed, but only three names are given. Clarify what each value corresponds to

**Language Note:** The review process has identified that the English language must be improved. PeerJ can provide language editing services - please contact us at [email protected] for pricing (be sure to provide your manuscript number and title). Alternatively, you should make your own arrangements to improve the language quality and provide details in your response letter. – PeerJ Staff

Reviewer 1 ·

Basic reporting

Although the paper highlights the enhanced recognition accuracy, the computational complexity and resource requirements are not sufficiently discussed. Given the use of a multi-layer CNN and the F-HRSF feature extraction, the model might require substantial computational resources.
The authors should include a more detailed analysis of the training time, memory usage, and inference time, especially in comparison with other benchmark methods (MFCC-LSTM or MFSC-CNN).
The incorporation of F-HRSF to weight subbands is an innovative approach, but the manuscript lacks a comprehensive explanation of the mathematical formulation for this transformation.

Experimental design

The authors should include a more detailed analysis of the training time, memory usage, and inference time, especially in comparison with other benchmark methods (MFCC-LSTM or MFSC-CNN).
The incorporation of F-HRSF to weight subbands is an innovative approach, but the manuscript lacks a comprehensive explanation of the mathematical formulation for this transformation.

Validity of the findings

Equations (5) and (6) introduce the concept of re-fitting the Half-Rising Sine Function (HRSF) based on Fisher ratios, but the specific procedure for this re-fitting is not sufficiently detailed.
How exactly is the sine function transformed? Is the Fisher ratio applied across all subbands or just specific ones, and how does this transformation affect the overall feature space?
The manuscript mentions that F-HRSF suppresses high and low-frequency subbands prone to noise. A clearer explanation of how the weighting of different frequency bands is achieved using the Fisher-enhanced HRSF (F-HRSF) would help.

Additional comments

Additionally, it would be helpful to compare the signal-to-noise ratio (SNR) or another quantifiable measure before and after applying F-HRSF to illustrate its effectiveness in noise suppression.
The paper uses accuracy, sensitivity, specificity, and the F-score for model evaluation, which are standard metrics. However, for a more comprehensive evaluation, it would be beneficial to also include additional metrics such as precision-recall curves, AUC (Area Under the Curve), and confusion matrix analysis. These metrics are critical in multi-class classification tasks, especially in distinguishing between less frequent classes.

Annotated reviews are not available for download in order to protect the identity of reviewers who chose to remain anonymous.

·

Basic reporting

Clarity and Language:
✅ Strength: The manuscript is generally well-written in formal and professional English. The language is technically sound and understandable to an academic audience.

⚠ Suggestion: Some minor grammatical inconsistencies and typographical redundancies appear occasionally (e.g., “inter-class scatter is the inter-class differentiation” is slightly repetitive). A professional language polish is recommended.

Structure and Referencing:
✅ Strength: The structure aligns with PeerJ standards: abstract, introduction, related work, methodology, experiments, and conclusions.

✅ Strength: Citations are relevant and recent (mostly from 2023–2024), providing a solid literature backdrop.

Experimental design

Scope and Novelty:
✅ Strength: The integration of Fisher Half-Rising Sine Function (F-HRSF) with MFCC and CNN is a novel approach in the context of piano audio recognition.

✅ Strength: The focus on capturing both static and dynamic features enhances relevance.

Methodology:
✅ Strength: The MFCC extraction, Fisher Ratio-based weighting, and CNN architecture are clearly explained with mathematical rigor.

⚠ Suggestion: The feature extraction equations are valid, but some equations lack explicit variable definitions (e.g., ka, kb, and x in Equation 5 could be better defined).

⚠ Suggestion: The dataset (GTZAN) is appropriate but not without known limitations. More discussion on the choice of dataset (e.g., genre duplication issues) is required.

✅ Strength: The inclusion of first-order differences in MFCCs to capture dynamic features is effective.

Validity of the findings

Results and Evaluation:
✅ Strength: Experiments are well-structured and use appropriate evaluation metrics: Accuracy, Sensitivity, Specificity, and F-score.

✅ Strength: The F-HRSF method outperforms benchmarks like MFSC-CNN and MFCC-LSTM, particularly on false negative reduction.

✅ Strength: Figures (3–7) support the narrative effectively, and the confusion matrix substantiates claims of high class-wise accuracy.

⚠ Limitation Noted: While MFSC-CNN outperforms in accuracy, the paper's focus on F-score justifies its claim of superiority.

Reproducibility:
⚠ Minor Weakness: Code availability or reproducibility scripts are not explicitly discussed. Including links to code repositories (e.g., GitHub) or data pipelines would improve transparency.

Additional comments

Strengths:
The approach is technically robust and well-articulated.

It fills a gap in MFCC-based piano feature extraction by addressing the lack of dynamic sensitivity.

The model architecture and performance metrics are clear and reasonable.

Areas for Improvement:
Dataset Limitations: The GTZAN dataset, while widely used, has known issues (repetition, imbalance). Discuss alternatives or future use of diverse datasets.

Model Interpretability: Like most deep models, this work suffers from low interpretability. A short discussion or use of tools like Grad-CAM or SHAP for visual interpretation would strengthen it.

Real-World Use: The paper lacks an analysis of real-time deployment feasibility. Latency, inference speed, and mobile deployment potential should be mentioned.

---

## Round 0.2 · accepted · Accept

Thank you for your resubmission. The reviewers have now commented and they are satisfied with your revised work. Therefore, I am pleased to inform you about the acceptance of your article. Thank you for your fine contribution.

Reviewer 1 ·

Basic reporting

The authors have addressed all the comments effectively in the revised version, so I am giving my consent that the paper is now fit for publication.

Experimental design

The authors have addressed all the comments effectively in the revised version, so I am giving my consent that the paper is now fit for publication.

Validity of the findings

The authors have addressed all the comments effectively in the revised version, so I am giving my consent that the paper is now fit for publication.

Additional comments

The authors have addressed all the comments effectively in the revised version, so I am giving my consent that the paper is now fit for publication.

·

Basic reporting

The manuscript is written in clear and professional English, with appropriate academic style and tone. The introduction establishes the motivation well and provides sufficient background with relevant references. The manuscript is logically structured, and figures are clear and informative. The raw data and code references appear adequate, and figures show no evidence of inappropriate manipulation. Overall, the reporting quality fully meets PeerJ standards.

Experimental design

The study is well within the journal’s scope as an AI Application article. The research question is clearly stated and meaningful, focusing on improving MFCC-based feature extraction for piano audio analysis. The methodology is rigorous and described in sufficient detail to allow replication, including dataset details (GTZAN), preprocessing steps, feature extraction procedures, and CNN architecture. The computational setup and software environment are also documented. Ethical considerations are respected, as no sensitive or human-subject data are involved.

Validity of the findings

The experiments are well designed and evaluated using multiple performance metrics (accuracy, sensitivity, specificity, F-measure). Comparisons with baseline methods (DTW, MFSC-CNN, MFCC-LSTM) strengthen the validity of the findings. Results are consistent and reproducible, and the discussion appropriately addresses robustness, generalization, and noise resilience. The limitations are acknowledged, particularly dataset generalizability and model interpretability. The conclusions are supported by the presented results.

Additional comments

This is a strong manuscript with clear contributions:

An innovative integration of Fisher-enhanced HRSF with MFCC for subband weighting.

A multilayer CNN model that demonstrates competitive and robust performance in piano audio recognition.

Thorough experimental validation, including comparisons, robustness tests, and interpretability analyses.

The manuscript is well-prepared, revisions have effectively addressed earlier concerns, and the study provides valuable insights for music information retrieval research.

I recommend acceptance without further revisions.